# Eucalyptus Leaf Solution to Replace Metals in the Removal of Cyanobacteria in Wastewater from the Paper Mill Industry

Zhewei Hu [1], Shu Jin [1,*], Rongrong Ying [1,*], Xiaohui Yang [2] and Baoping Sun [3]

1   Nanjing Institute of Environmental Sciences, Ministry of Ecology and Environment of the People's Republic of China, Nanjing 210042, China; huzhewei@nies.org
2   Chinese Academy of Forestry, Beijing 100864, China; xiaohuiyang@caf.ac.cn
3   School of Soil and Water Conservation, Beijing Forestry University, Beijing 100083, China; sunbaoping@bjfu.edu.cn
*   Correspondence: jsofrcc@126.com (S.J.); yingrongrong@nies.org (R.Y.)

**Abstract:** The frequent occurrence of cyanobacterial blooms, caused by the eutrophication of water bodies, has triggered several ecological issues. Metal-controlled cyanobacteria are resulting in a series of secondary environmental problems and thus limiting environmental sustainability. Whether there is a more environmentally friendly way to replace metals in the removal of cyanobacteria is still unclear. To explore whether common heavy metals inhibit algal growth and whether *Eucalyptus* leaves (EL) can replace heavy metal ions in controlling algae outbreaks, here, we add $Fe^{3+}$, $Al^{3+}$, 3 mol/L of zinc ($Zn_3$), 10 mol/L zinc ($Zn_{10}$), and EL to a medium containing *Cyanobacteria*. We determine the medium's color (456 nm), UV (254 nm), chlorophyll a, turbidity, temperature, pH, total dissolved solids, conductivity, and blue-green algae (BGA) at days 1, 4, 7, 11, 14, 19, and 21. We find that $Fe^{3+}$, $Al^{3+}$, $Zn_3$, $Zn_{10}$, and EL can inhibit chlorophyll synthesis, thereby impeding algae biomass growth due to metal ions' disruption of the chlorophyll structure. The toxicity of $Zn^{2+}$ may be higher than that of $Fe^{3+}$ and $Al^{3+}$ since it can completely destroy the structure of chlorophyll a. The damage of Zn (10) to chlorophyll a is stronger than that of Zn (3), indicating that high concentrations of metals have a stronger inhibitory effect on algae. The toxicity of EL to algae is lower than that of other metals, but it can significantly inhibit the growth of algae. We suggest the use of *Eucalyptus* leaves to inhibit algal growth in eutrophic water bodies. Our results provide a scientific basis for an environmentally friendly approach to controlling cyanobacteria outbreaks.

**Keywords:** metals; eucalyptus leaf; chlorophyll a; blue-green algae

## 1. Introduction

Eutrophication is caused by excessive levels of nutrients, such as nitrogen or phosphorus, under the influence of natural factors and human activities [1–4]. The frequent occurrence of cyanobacterial blooms, caused by the eutrophication of water bodies, has triggered a series of environmental and ecological issues that have recently attracted increased interest from the scientific community [5]. The methods of controlling water eutrophication are mainly divided into exogenous and endogenous controls [2]. There are several physical, biological, and chemical approaches and technologies to tackling cyanobacterial outbreaks [6–9]. However, although the physical and biological methods are environmentally friendly, they require large amounts of labor and resources, making them impractical [10,11]. Chemical methods, such as the use of metals, are highly effective in controlling cyanobacterial outbreaks, but can also cause secondary environmental problems [12–14]. In addition, metals can impact algal growth and reproduction. Previous studies have shown that when metals enter the water environment, even low concentrations of Zn and other metals can promote the growth and reproduction of algae cells since metal ions are essential for cell synthesis [15–18]. However, at excessive concentrations,

they inhibit algal growth. Under metal stress, algal synthesis will be affected at different degrees, depending on the metal itself, on its concentration, and on the algae species [19–21]. For example, when blue-green algae cells are stressed by metal ions, corresponding changes will appear on the cell surface. Vivacqua et al. showed that when *Chlorella vulgaris* was under copper stress, the cells would dissolve and break up [22]. Similarly, Coruh et al. reported that the chlorophyll a content of algae gradually decreased with increasing $Zn^{2+}$ levels [23]. Zhang et al. found that when the concentration of $Al^{3+}$ was 1mg/L and 5 mg/L, the content of chlorophyll-a in *Chlorella vulgaris* was almost unaffected, while the synthesis of chlorophyll-a was greatly inhibited by 10 mg/L, 50mg/L, and 80mg/L of $Al^{3+}$ [17]. Heavy metals can destroy the protein structure or replace elements by combining with the sulfhydryl group in a protein, thus causing toxic effects on algae [20]. In addition, $Zn^{2+}$ has stronger toxic effects than other metals, thus inhibiting the chlorophyll fluorescence of some algae [24]. Some metals can inhibit the growth and metabolism of algae cells by preventing cell division, inhibiting the synthesis of cytochrome, affecting photosynthesis, changing the genetic material, causing the distortion of algae cells, and changing the species composition of algae in the natural environment [25]. It was found that the photosynthetic electron transport chain of Photosynthetic system II(PSII) of *Scenedesmus obliquus* treated with a high concentration of $Fe^{2+}$ was significantly affected, which reduced the light energy utilization efficiency and photosynthetic efficiency [19].

Although metals can significantly inhibit algae reproduction and outbreaks, their impacts on human health and the natural environment should not be neglected [26]. Most heavy metal pollution is usually related to toxicity, especially when metals exist in a dissolved form [4]. Metals can enter the human body through inhalation, water, diet, and skin exposure, causing chronic or acute poisoning [27]. Aquatic plants are also widely used to remove excessive nutrients in water. However, some places are not suitable for planting aquatic plants. In this sense, other substances should be used to control cyanobacteria outbreaks.

Previous studies have shown that *Eucalyptus* leaf extract has a strong inhibitory effect on the root and stem length of sorghum, wheat, cucumber, and rape seedlings [28–31]. Based on this, we speculate that the color of *Eucalyptus* leaves, indicating their organic matter content and micro-toxicity, may be related to algae photosynthesis, potentially inhibiting algae outbreaks. Therefore, we added $Fe^{3+}$, $Al^{3+}$, 3 mol/L of zinc ($Zn_3$), 10 mol/L zinc ($Zn_{10}$), and EL to a medium containing *Cyanobacteria* and determined the growth index of algae during the culture period. The objectives of this study were to: (i) explore whether the common metals of $Fe^{3+}$, $Al^{3+}$, and $Zn^{2+}$, along with *Eucalyptus* leaves, inhibit algal growth; (ii) explore whether *Eucalyptus* leaves can replace metal ions in controlling algae outbreaks. Our results provide a scientific basis for an environmentally friendly approach to controlling cyanobacteria outbreaks.

## 2. Materials and Methods

### 2.1. Experimental Design

The algae species selected for the experiment were cyanobacteria, obtained from the wastewater sedimentation tank of a closed paper mill in Hefei, Anhui Province, China. The wastewater was light green. The pH and temperature of the wastewater were 7.3 and 22.3 °C. Eighteen samples (of about 1800 mL each) were randomly collected from the wastewater precipitation tank and poured into 2500-mL glass beakers. The samples were randomly divided into groups A, B, C, D, E, and F, with three samples in each group (Table 1), and cultured in an American Society for Microbiology (ASM) medium under aseptic conditions. The composition of the ASM culture medium is shown in Table S1. Prior to culturing, the culture medium and containers were sterilized at 121 °C for 30 min.

**Table 1.** Variations in chlorophyll a increment, respectively. The A, B, C, D, E, and F treatments underwent the addition of $Fe^{3+}$, $Al^{3+}$, 3 mol/L zinc ($Zn_3$), 10 mol/L zinc ($Zn_{10}$), *Eucalyptus* leaves (EL), and pure water, respectively. There were significant differences among treatments and culture days, respectively ($p < 0.05$).

| Cultivation Days | A | B | C | D | E | F |
|---|---|---|---|---|---|---|
| 1 | $-11.62 \pm 2.93$ | $-4.51 \pm 0.50$ | $-6.29 \pm 0.02$ | $-14.97 \pm 1.20$ | $84.73 \pm 2.15$ | $-2.47 \pm 0.61$ |
| 4 | $-8.14 \pm 2.33$ | $-2.39 \pm 0.86$ | $-9.65 \pm 0.37$ | $-13.42 \pm 1.79$ | $-88.86 \pm 2.80$ | $5.21 \pm 0.61$ |
| 7 | $-1.91 \pm 0.37$ | $-3.31 \pm 0.67$ | $-2.38 \pm 0.25$ | $-0.17 \pm 0.04$ | $-14.05 \pm 2.24$ | $7.93 \pm 1.05$ |
| 11 | $-0.75 \pm 0.01$ | $-1.98 \pm 0.11$ | $0.49 \pm 0.13$ | $-0.12 \pm 0.06$ | $-5.9 \pm 1.73$ | $3.95 \pm 0.51$ |
| 14 | $-1.24 \pm 0.72$ | $0.12 \pm 0.06$ | $1.85 \pm 0.58$ | $0$ | $-3.47 \pm 0.03$ | $-4.66 \pm 0.78$ |
| 19 | $-0.02 \pm 0.01$ | $-0.04 \pm 0.02$ | $3.25 \pm 0.29$ | $0$ | $-0.1 \pm 0.07$ | $-0.31 \pm 0.01$ |
| 21 | $-1.68 \pm 0.21$ | $-3.91 \pm 0.15$ | $-5.75 \pm 0.36$ | $0$ | $-8.86 \pm 1.85$ | $-26.32 \pm 1.21$ |

After transportation to the laboratory, the algae were cultured under a light intensity of about 3000 lx. The dark period of light illumination was t (bright): t (dark) = 12 h: 12 h, and the temperature was 25 °C. Artificial oscillation was carried out daily to ensure the normal growth of blue-green algae.

Subsequently, 650 mL of $Fe^{3+}$, $Al^{3+}$, 3 mol/L zinc ($Zn_3$), 10 mol/L zinc ($Zn_{10}$), and *Eucalyptus* leaves (EL) were added; as a control (CK), we used 650 mL of water without any metals (Figure 1). The algae species were cultured for 21 days, and the color (456 nm), UV (254 nm), chlorophyll a, turbidity, temperature, pH, total dissolved solids, conductivity, and blue-green algae (BGA) in the sample bottles were determined at days 1, 4, 7, 11, 14, 19, and 21.

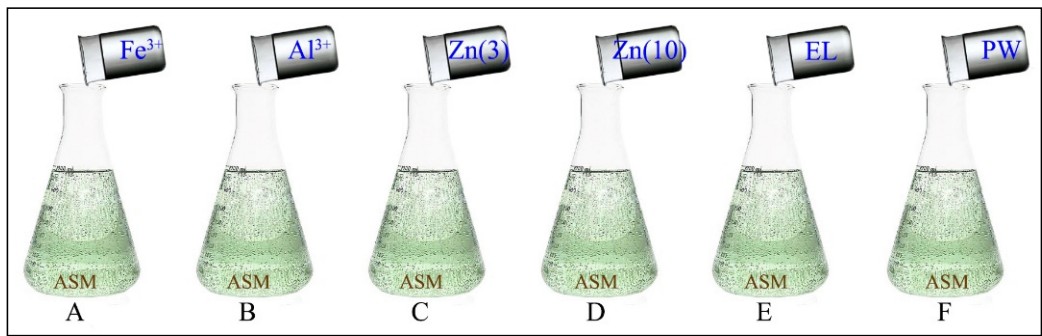

**Figure 1.** Schematic diagram of the six treatments. The samples were randomly divided into groups A, B, C, D, E, and F (light green). An American Society for Microbiology medium (ASM) was added to the A, B, C, D, E, and F treatments, along with $Fe^{3+}$, $Al^{3+}$, 3 mol/L zinc Zn (3), 10 mol/L zinc Zn (10), *Eucalyptus* leaves (EL), and pure water, respectively.

*2.2. Index Measurement and Analysis*

2.2.1. Preparation of the *Eucalyptus* Leaf Solution

Fresh *Eucalyptus* leaves were washed with distilled water, cut into pieces, and placed into a 1-L flask drying, followed by the addition of three times the volume of methanol. Extraction was performed three times (2 h each) by ultrasonic waves at 25 °C. The three filtrates were combined and concentrated under reduced pressure to obtain the extract, which was stored at 4 °C until use.

2.2.2. Determination of the pH and Temperature of Water

Water temperature and pH were measured using a multifunctional water quality detector. We rinsed the probe with clean water and dried it with a paper towel. Then, we inserted the probe in the sample and recorded pH and temperature readings, repeating this three times for each sample. After measuring a sample, we rinsed the probe with clean water and dried it with a clean paper towel.

### 2.2.3. Determination of Chlorophyll a Concentration

We used the hot ethanol method for the measurement of chlorophyll a concentrations. First, 25 mL of the shaken algal fluid was taken via a pipette and suction-filtered by 45-μm glass fiber filters. Second, the samples, together with the filter membrane sheared into fine pieces, were placed in a 10-mL colorimetric tube, followed by the addition of 10 mL of 95% ethanol and extraction in a water bath at 75 °C for 5 min. All samples were then uniformly placed in a 4 °C freezer for 12 h to extract chlorophyll a. Absorbance was measured at 649 and 665 nm. The chlorophyll a content was calculated as follows:

$$X \ (ug/L) = 13.95 A_{665} - 6.88 A_{649} \tag{1}$$

where $A_{665}$ and $A_{649}$ are the absorbance values of the algae solution at 665 and 649 nm, respectively.

### 2.2.4. Determination of Turbidity

For turbidity analysis, we used the equation constructed by Púrtielje, based on measurements in shallow lakes in the Netherlands [11]. The minimum turbidity value (i.e., the maximum $Z_{SD}$) of 0.16/m in the dataset was set as $Turb$Bck, assuming that there were no planktonic algae and no suspended algae particles at this time. In addition, we established the empirical relationship between chlorophyll a and $Turb_{Tot}$ ($1/Z_{SD}$). The chlorophyll a concentration had a corresponding minimum $1/Z_{SD}$ value. The slope of this line was taken as 5% of ($Turb_{Tot} - Turb_{Bck}$)/chlorophyll-a, and the result was 0.01/m. Therefore, to calculate algae turbidity, we used the following equation:

$$Turb = 1/Z_{SD} - 0.16 - 0.01 \ \text{chlorophyll a} \tag{2}$$

### 2.2.5. Determination of BGA

We used the turbidimetric method and the dry weight method to measure algae biomass. After the cyanobacteria were continuously cultured for six days, they were diverted into nine samples with different concentrations, according to a certain proportion. The absorbance values of each concentration were measured at a wavelength of 456.0 nm, and the samples were dried to a constant weight to obtain the dry cell weight (mg/L).

### 2.3. Data Analysis

All statistical analyses were performed using the software package SPSS 16.0. Descriptive statistics were applied to calculate the means and standard deviations for each set of replicates. A two-way ANOVA was used to analyze the differences in color, UV, chlorophyll a, turbidity, temperature, pH, total dissolved solids, conductivity, and BGA, using the treatment and culture days as independent factors. In addition, we determined the correlations between BGA, total dissolved solids (TDS), and temperature, as well as chlorophyll a, turbidity, and conductivity. All data met the requirements of a normal distribution and homogeneity of variance.

## 3. Results

### 3.1. Variations in Water's Physicochemical Properties between Treatments

The pH and temperature of water can reflect algal growth [32]. Via photosynthesis, algae can increase the dissolved oxygen content in the water, resulting in a higher pH [33]. In addition, the energy released through algae cell reproduction can also increase the water temperature [34]. Here, the pH of the culture medium increased first and then decreased over time (Figure 2). The pH of CK fluctuated greatly, with an average of 8.7, which was significantly higher than that of the other treatments ($p < 0.05$); in CK, it reached the maximum value of 10.3 on day 7. The pH fluctuation of Zn (10) was relatively low, with an average of 7.34, registering as 19.1% lower than that of CK. The control had the lowest pH of 6.9, which was significantly lower than that of the other treatments ($p < 0.05$). The pH of Zn (10) was significantly higher than that of Zn (3), but significantly lower than that of EL

($p < 0.05$). In addition, the pH values of $Fe^{3+}$ and $Al^{3+}$ were significantly lower than those of the other treatments. Based on these results, we infer that algal growth was highest in CK and lowest in EL. This finding is in agreement with a previous study [35].

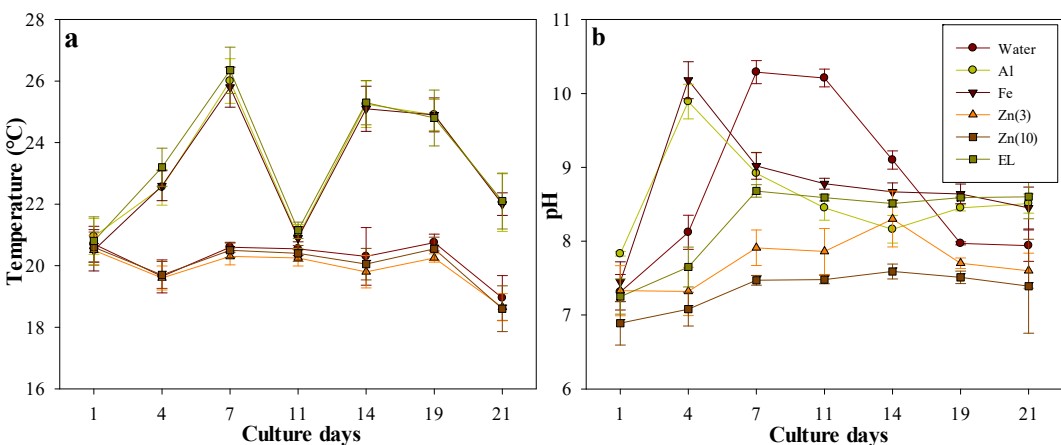

**Figure 2.** Variations in mean temperature and pH ($\pm$S.D.) of water in the different treatments over a period of 21 days (**a** is temperature, **b** is pH).

The temperature of the culture medium showed two different trends with the addition of different inhibitors (Figure 2). The temperature change trends of CK, Zn (3), and Zn (10) were similar, whereas those of the EL, $Fe^{3+}$, and $Al^{3+}$ treatments were similar. The temperatures of the EL, $Fe^{3+}$, and $Al^{3+}$ treatments decreased in the 11 days, maybe as the algae suspended growth during this period. This can also be confirmed by the biomass data of cyanobacteria. The temperatures of the treatments EL, $Fe^{3+}$, and $Al^{3+}$ were significantly higher than those of the CK, Zn (3), and Zn (10) treatments. In EL treatments, the temperature reached a maximum value of 36.35 °C on day 7, indicating that the *Eucalyptus* leaves released more energy during this period. The lowest temperature of Zn (3) was 19.60 °C on day 3, and the maximum value was 85% higher than the minimum value. It was found that temperature had a significant effect on the BGA growth and chlorophyll-a synthesis of EL [18]. Most likely, variation in temperature affected the photosynthesis and respiration intensity of algae, with subsequent impacts on growth and development [17].

### 3.2. Variations in Color and UV

The color and UV of the culture medium could reflect algal growth and concentration [36]; the darker the color, the more pronounced was the growth of algae [37]. The color and UV of different treatments gradually decreased over time, except for the control (Figure 3). This indicated that metal ions and *Eucalyptus* leaves inhibited algae reproduction. We observed considerable differences between EL and the other treatments, most likely because *Eucalyptus* leaves contain more chloroplasts. The average color and UV values of EL were 755.50 and 1.51, respectively, which were higher than those of the other treatments. It is possible that the EL contained chlorophyll, resulting in higher color and UV values. In the early stage of the culture, the algae propagated in large quantities by using the nutrients in *Eucalyptus* leaves and the culture medium, and reached the maximum value of propagation. However, when the nutrients were exhausted, the number and growth of algae would slow down, resulting in decreases in color and UV. However, in other treatments, metal ions always affected the growth and reproduction of algae. The maximum color and UV values for EL treatment were 1025.70 and 2.01, respectively, on day 1, reaching 501.07 and 1.07, respectively, on day 21.

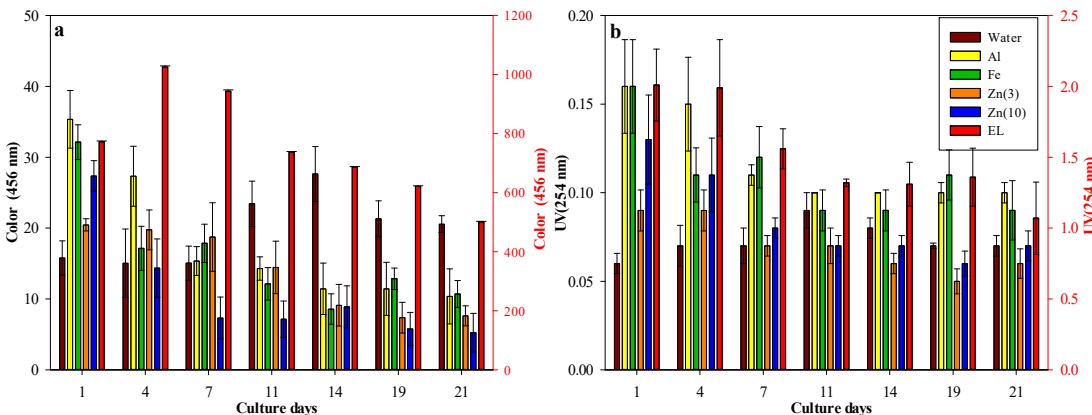

**Figure 3.** Variations in color and UV values (±S.D.) in the different treatments (**a** is color, **b** is UV).

### 3.3. Variations in Water Conductivity

Conductivity can reflect the decomposition or destruction of algae [38]. The higher the degree of damage, the more ions are released, resulting in higher water conductivity [2]. The control showed an average conductivity of 393.28 us/cm, with no significant changes over time ($p > 0.05$). The conductivity of the metal ion and EL treatments decreased over time (Figure 4). In EL, the average conductivity was highest, with 954.87 us/cm, whereas in CK, it was 41% lower. In addition, the conductivity levels of the $Fe^{3+}$ and $Al^{3+}$ treatments were significantly higher than those of the CK, Zn (3), and Zn (10) treatments. This may be related to the enrichment efficiency of metal ions and algae cells and the toxicity of metals. When metal ions enter the water and come into contact with algae, those with positive charges will bind to the negatively charged functional groups on the cell walls, thereby accumulating on the cell surface [39]. The ability of different metal ions to accumulate on algal cell surfaces is related to electrostatic attraction and the radii of hydrated ions [40]. The EL treatment had the highest water conductivity of 1113.95 us/cm on day 1 and the lowest of 892.60 us/cm on day 21. The conductivity of EL was higher than that of the treatments with metals, most likely because *Eucalyptus* leaves contain more ions.

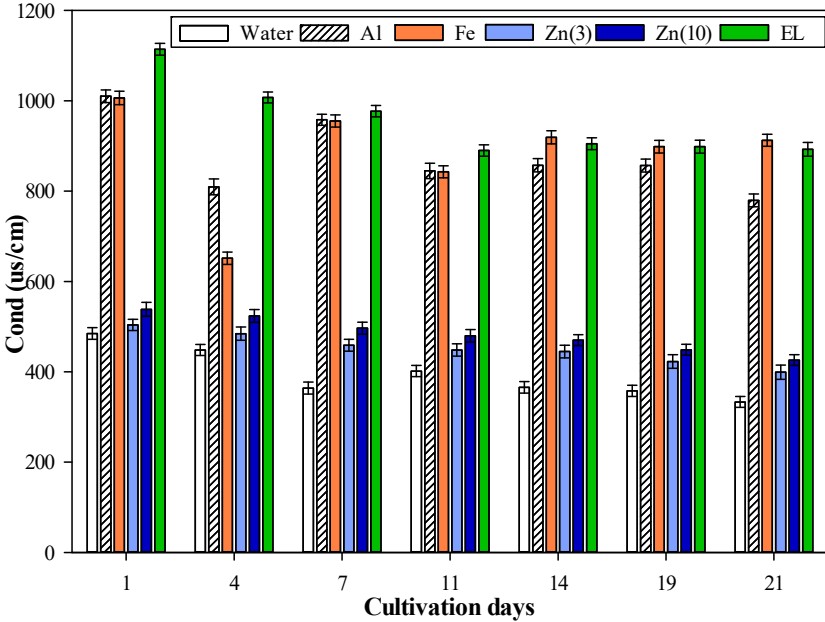

**Figure 4.** Variations in mean (±S.D.) conductivity of the water in the different treatments.

### 3.4. Variations in Total Dissolved Solids and Turbidity

There were significant differences in TDS and $Turb_{NTU}$ among the six treatments (Figure 5). For EL, CK, Zn (3), and Zn (10), we observed a downward trend in the number of cultivation days. Although there were no significant changes in the $Fe^{3+}$ and $Al^{3+}$ treatments in the first 11 days, the levels decreased in the last 10 days. The TDS of treatment EL was highest, with an average of 636.71mg/L, which was 1.2 times that of CK. The TDS value of the EL treatment was highest (752.50 mg/L) on day 1 and reached a minimum of 580.50 mg/L on day 21. The turbidity of the Zn (3) and Zn (10) treatments decreased over time, and other treatments showed a trend of increase, followed by decrease. The average turbidity of CK was 68.62 NTU (%), which was significantly higher than that of the other treatments. The turbidity levels of Zn (3), Zn (10), and EL tended to be consistent and close to zero, indicating that metals inhibited algal growth, although *Eucalyptus* leaves had a stronger effect. It was found that different concentrations of $Fe^{3+}$, $Al^{3+}$, and $Zn^{2+}$ could promote the growth and reproduction of *Cyanobacteria* in a certain concentration range [41]. $Fe^{3+}$, $Al^{3+}$, and $Zn^{2+}$ could inhibit algal growth at a high concentration [40]. However, in another study, the toxic effect of $Al^{3+}$ was greater than that of $Zn^{2+}$ [42,43]. The toxicity of metals may also be related to the tolerance of algae species to different metals. In our study, $Zn^{2+}$ was more toxic to *Cyanobacteria* cells under the long-term stress of $Fe^{3+}$, $Al^{3+}$, and $Zn^{2+}$ since the TDS and Turb of $Zn^{2+}$ were lower than those of $Fe^{3+}$ and $Al^{3+}$. In addition, *Eucalyptus* leaves clearly inhibited algal growth, albeit only to a slightly lower extent than metal ions.

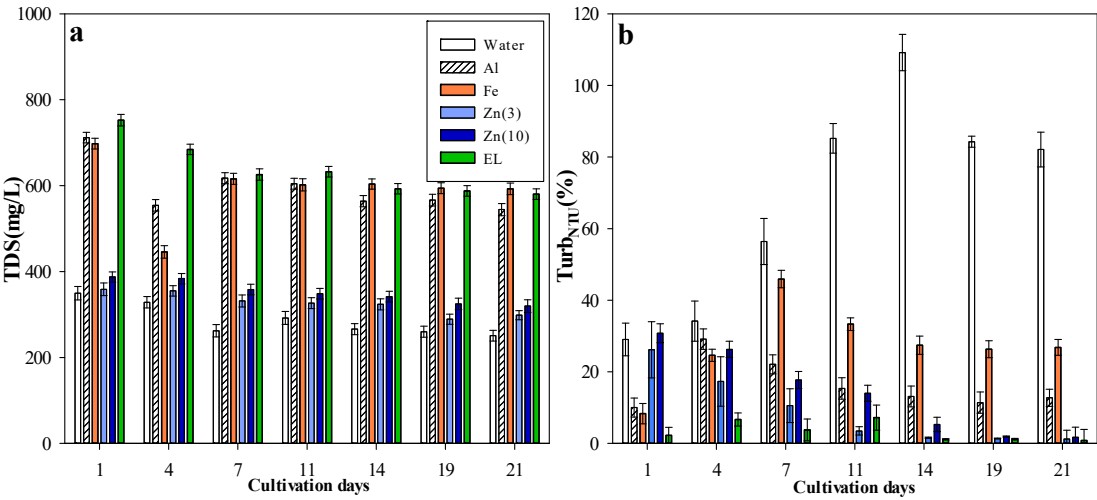

**Figure 5.** Variations in mean (±S.D.) total dissolved nitrogen and turbidity of water in the different treatments. TDS and $Turb_{NTU}$ color (456 nm) were the total dissolved solids and turbidity NTU in 456 nm, respectively (**a** is TDS, **b** is Turb).

### 3.5. Variations in Chlorophyll a and Blue-Green Algae

There was a significant difference between the treatments and CK ($p < 0.05$) (Table 2). Different concentrations of $Fe^{3+}$, $Al^{3+}$, $Zn^{2+}$, and EL had different effects on the content of chlorophyll a in *Cyanobacteria* cells. However, the changing trend of chlorophyll a was similar to that of BGA. This was because high concentrations of chlorophyll a could effectively promote BGA. The levels of chlorophyll a and BGA in CK first increased and then stabilized, whereas chlorophyll a and BGA decreased over time in the treatments $Fe^{3+}$, $Al^{3+}$, Zn (3), and Zn (10). Especially in Zn (10), the chlorophyll a and BGA levels reached zero on day 14. The average chlorophyll a and BGA levels of EL were 34.10 and 13.70 ug/L, respectively, which were significantly higher than those in the treatments with metal ions. Compared with metal treatments, the chlorophyll a and BGA levels of EL decreased more slowly, most likely because of the slow decomposition of *Eucalyptus* leaves.

**Table 2.** Variations in chlorophyll a increment, respectively. To the A, B, C, D, E, and F treatments we added $Fe^{3+}$, $Al^{3+}$, 3 mol/L zinc ($Zn_3$), 10 mol/L zinc ($Zn_{10}$), *Eucalyptus* leaves (EL), and pure water, respectively. There were significant differences among treatments and culture days, respectively ($p < 0.05$).

| Cultivation (days) | A | B | C | D | E | F |
|---|---|---|---|---|---|---|
| 1 | $-18.69 \pm 2.82$ | $-25.68 \pm 1.53$ | $-11.81 \pm 24.84$ | $-23.91 \pm 2.22$ | $-12.52 \pm 2.34$ | $-1.72 \pm 0.22$ |
| 4 | $-20.94 \pm 3.42$ | $-12.67 \pm 1.27$ | $-35.9 \pm 25.14$ | $-59.99 \pm 1.8$ | $-16.09 \pm 2.46$ | $13.28 \pm 2.04$ |
| 7 | $-4.09 \pm 0.71$ | $-7.67 \pm 1.95$ | $-12.54 \pm 7.26$ | $-0.58 \pm 0.02$ | $3.48 \pm 1.52$ | $20.55 \pm 2.65$ |
| 11 | $-1.04 \pm 0.17$ | $-1.6 \pm 0.03$ | $-0.09 \pm 0.055$ | $-0.06 \pm 0.01$ | $-14.42 \pm 1.52$ | $12.43 \pm 1.68$ |
| 14 | $-0.23 \pm 0.12$ | $-0.26 \pm 0.05$ | $-0.05 \pm 0.01$ | $0$ | $-0.66 \pm 0.03$ | $-23.35 \pm 2.8$ |
| 19 | $0.19 \pm 0.03$ | $0$ | $0$ | $0$ | $-0.03 \pm 0.01$ | $-1.09 \pm 0.17$ |
| 21 | $-0.33 \pm 0.04$ | $0$ | $0$ | $0$ | $-0.07 \pm 0.02$ | $0$ |

Previous studies have shown that photosystem II is the main target of $Fe^{3+}$ and $Al^{3+}$, and high concentrations of these ions will have a strong and negative impact on the photosynthetic electron transport chain [44]. In addition, $Fe^{3+}$ and $Al^{3+}$ can seriously damage the chloroplast structure and form coordination compounds with chlorophyll, thereby impeding the formation and stability of chlorophyll and reducing the content of chlorophyll, which results in cell chlorosis and discoloration [45]. The content of chlorophyll a was significantly inhibited at the concentrations of 3 and 10 mol/L of $Zn^{2+}$, most likely because $Zn^{2+}$ combines with the thiol groups of prochlorophyll ester reductase, δ-aminolevulinic acid synthetase, and bilinogen deaminase after entering the cell, changing the composition of enzyme molecules and reducing enzyme activity [46]. Similar to metals, high concentrations of *Eucalyptus* leaf solution in the cell will damage the function of the chloroplast and change the composition of chloroplast proteins, thus hindering the synthesis of chlorophyll a (Figure 6).

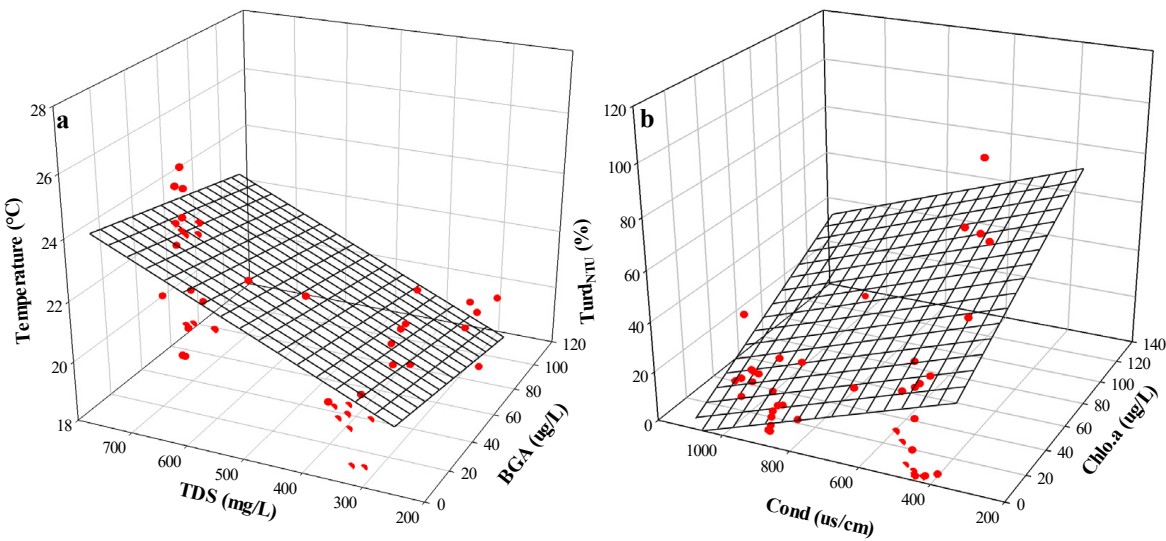

**Figure 6.** Effects of various inhibitors on algal growth. (**a**) represents the relationships among temperature, total dissolved nitrogen, and blue-green algae, and (**b**) the relationship among the turbidity NTU, chlorophyll a, and conductivity.

## 4. Conclusions

To determine whether common metals inhibit algal growth and whether EL can replace metal ions in controlling algae outbreaks, we added $Fe^{3+}$, $Al^{3+}$, Zn (3), Zn (10), and EL to an ASM medium, measuring changes in color, UV, chlorophyll a, turbidity, temperature, pH, total dissolved solids, conductivity, and BGA. The pH values of the treatments with $Fe^{3+}$, $Al^{3+}$, Zn (3), Zn (10), and EL increased significantly and were higher than that of CK. The average color and UV values of EL were 755.50 and 1.51, respectively,

which were higher than those of the other treatments. The TDS of the EL treatment was highest, with an average of 636.71mg/L, which was 1.2 times that of CK. $Fe^{3+}$, $Al^{3+}$, Zn (3), Zn (10), and EL could inhibit the synthesis of chlorophyll and not affect the algae biomass. In addition, it was found that the toxicity of $Zn^{2+}$ may be higher than that of $Fe^{3+}$ and $Al^{3+}$ since it can completely destroy the structure of chlorophyll a. Based on our results, *Eucalyptus* leaves can be effectively used to inhibit algal growth in eutrophic water bodies.

**Supplementary Materials:** The following are available online at https://www.mdpi.com/article/10.3390/w13081014/s1, Table S1: Composition of the ASM (American Society for Microbiology) culture medium.

**Author Contributions:** Z.H. wrote the paper; S.J. conceived and designed the experiments; R.Y. and X.Y. analyzed the data; B.S. performed the experiments and collected the data. All authors have read and agreed to the published version of the manuscript.

**Funding:** This research was funded by the National Natural Science Foundation of China, grant number 42007182.

**Institutional Review Board Statement:** Not applicable.

**Informed Consent Statement:** Not applicable.

**Data Availability Statement:** The data that support the findings of this study are available upon request from the authors.

**Conflicts of Interest:** The authors declare no conflict of interest.

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
