# Peer review of "Eucalyptus Leaf Solution to Replace Metals in the Removal of Cyanobacteria in Wastewater from the Paper Mill Industry"

_water, doi:10.3390/w13081014_

Round 1

Reviewer 1 Report

Dear Authors,

The background of this work is interesting. Explore the heavy metal inhibit algae growth can be has potential, in special associated to water contamination. But there are significant deficiencies in the manuscript. For example, the authors confuse basic soil concept because elements as Fe or Al are not heavy metals.

The results section does not present a discussion of the data or comparison with other studies. The submitted manuscript concludes that Eucalyptus leaves can replace heavy metal ions in controlling algae outbreaks. You should discuss this better, It is important.

Furthermore, results presentation and the statistical analysis is very poor to study the relationship

Specific comments:

-Title: change the title, do not use the term “heavy metals” because really you have only evaluated Zn, as I have commented Fe or Al are not heavy metals

-Abstract: The abstract needs rewriting with the objective of the work. The abstract must be structured so that it starts by explaining the problems and the background, what has been done, and what is new in this study, to then finally present the study objective. For example delete “The pH levels in 16 the treatments containing Fe3+, Al3+, Zn3, Zn10, and EL increased significantly and were higher than 17 that of the control” in Line 18 and 19 “We found that Fe3+, Al3+, Zn3, Zn10, and EL can inhibit chlorophyll synthesis, thereby impeding algae biomass growth”, explain why

-In the introduction it is necessary to explain the importance of the elements in waters and the relationship with environmental problems, which elements are problematic and which are the natural levels, or what concentration by other studies. For example in lines 39-40 “low concentrations of Cu, Zn, Pb, Ni, Cd, Cr, and other heavy metals can promote the growth and reproduction of algae cells since heavy metal ions are essential for cell synthesis”. You don't analyze Cu, Pb, Ni, Cd, Cr; why then do you comment on this in the introduction? I recommend you focus on your study.

-Materials and methods:

1) Table 1: I think this table should appear as supplementary material

2) in section: 2.2.2. Determination of the Physicochemical Properties of Water, you only comment “Water temperature and pH were measured using a multifunctional water quality detector (Gao et al., 2016). There are many other physicochemical properties of water, if you only evaluate pH and temperature, I think it is not necessary to make a section for this.

3) Statistical analysis is very poor. Basically, it consists of one paragraph with general comments. Two-way ANOVA was used but if you have to make a table with the basic statistics for metals (mean, standard deviation, percentiles, maxims, .....) This will be easier to interpret, compare and discuss. On the other hand, I recommend that you use a statistical analysis based on mixed models, this is more original.

- Results and Discussion:

This is possibly the most important section of any scientific work and it should contain a discussion in accordance with the results obtained in the work itself and in published works by other authors. In this manuscript there is no discussion section. The author present is an exposition of raw analytical results.

In general, this section must be completely rewritten. Tables with important data from the study should be included and the new statistical analyses.

In general all the figures are of poor scientific quality, these look like a school work, I suggest you:

Figure 1: this has to be improved and What happened on day 11 at the temperature?

Join Figures 3, 4 and 5. This new has to be improved. Choose a more expressive type of graphic and use colors that are more distinctive

Figure 6. This has to be displayed as a table, and this table has to be discussed in depth

Author Response

Dear Editors and Reviewers:

Thank you for your letter and for the reviewers' comments concerning our manuscript entitled “Eucalyptus Leaf Solution Replace Metals to Removal of Cyanobacteria in Paper Mill Industry Wastewater” (No. water-1149172). Those comments are all valuable and very helpful for revising and improving our paper, as well as the important guiding significance to our researches. We have studied comments carefully and have made correction which we hope meet with approval. Besides, our current manuscript followed the journal formatting guidelines of Water. Revised portion are marked in red throughout the revised manuscript. The main corrections in the paper and the Responses to the reviewer’s comments are as flowing:

NOTE: All the Page and Line numbers where revisions were made refer to the Manuscript and Highlight with marked changes (Manuscript_revised version).docx. The Manuscript_Clean Version was the same version of the Manuscript_revised version with cleaned from all the marks.

Responses to the Reviewer #1’s comments:

  • The background of this work is interesting. Explore the heavy metal inhibit algae growth can be has potential, in special associated to water contamination. But there are significant deficiencies in the manuscript. For example, the authors confuse basic soil concept because elements as Fe or Al are not heavy metals.

Response: Thank you very much for your comments and suggestions to revised the manuscript. We revise the “heavy metal” to “metal” throughout the manuscript in the latest version of the manuscript.

  • The results section does not present a discussion of the data or comparison with other studies. The submitted manuscript concludes that Eucalyptus leaves can replace heavy metal ions in controlling algae outbreaks. You should discuss this better, It is important.

Response: Thank you very much for your and comments. We revised it and rewritten this part in the latest version of the manuscript.

  • Furthermore, results presentation and the statistical analysis is very poor to study the relationship

Response: Thank you very much for your and comments. We revised it and rewritten this part in the latest version of the manuscript.

  • Title: change the title, do not use the term “heavy metals” because really you have only evaluated Zn, as I have commented Fe or Al are not heavy metals

Response: Thank you very much for your and suggestions. We revised the “heavy metal” to “metal” in the title of the latest version of the manuscript (Page 1).

  • -Abstract: The abstract needs rewriting with the objective of the work. The abstract must be structured so that it starts by explaining the problems and the background, what has been done, and what is new in this study, to then finally present the study objective. For example delete “The pH levels in 16 the treatments containing Fe3+, Al3+, Zn3, Zn10, and EL increased significantly and were higher than 17 that of the control” in Line 18 and 19 “We found that Fe3+, Al3+, Zn3, Zn10, and EL can inhibit chlorophyll synthesis, thereby impeding algae biomass growth”, explain why

Response: Thank you very much for your and comments. We revised the abstract in the latest version of the manuscript (Page 1).

  • In the introduction it is necessary to explain the importance of the elements in waters and the relationship with environmental problems, which elements are problematic and which are the natural levels, or what concentration by other studies. For example in lines 39-40 “low concentrations of Cu, Zn, Pb, Ni, Cd, Cr, and other heavy metals can promote the growth and reproduction of algae cells since heavy metal ions are essential for cell synthesis”. You don't analyze Cu, Pb, Ni, Cd, Cr; why then do you comment on this in the introduction? I recommend you focus on your study.

Response: Thank you very much for your and comments. We revised the introduction in the latest version of the manuscript (Page 1-2).

  • Table 1: I think this table should appear as supplementary material.

Response: Thank you very much for your and comments. We revised the table and put it after the text in the latest version of the manuscript (Page 13).

  • in section: 2.2.2. Determination of the Physicochemical Properties of Water, you only comment “Water temperature and pH were measured using a multifunctional water quality detector (Gao et al., 2016). There are many other physicochemical properties of water, if you only evaluate pH and temperature, I think it is not necessary to make a section for this.

Response: Thank you very much for your and comments. We added the details in the latest version of the manuscript (Page 3-4).

  • This is possibly the most important section of any scientific work and it should contain a discussion in accordance with the results obtained in the work itself and in published works by other authors. In this manuscript there is no discussion section. The author present is an exposition of raw analytical results.

Response: Thank you very much for your and comments. We revised it and rewritten this part. We put the result analysis and discussion together in the latest version of the manuscript (Page 4-10).

  • In general, this section must be completely rewritten. Tables with important data from the study should be included and the new statistical analyses.

Response: Thank you very much for your and comments. We revised it and rewritten this part, we think that graph can better show data, so we still use graph to show data in the latest version of the manuscript (Page 4-10).

  • Figure 1: this has to be improved and What happened on day 11 at the temperature?

Response: Thank you very much for your and comments. We improved it and explained what happened on day 11 in the latest version of the manuscript (Page 5).

  • Join Figures 3, 4 and 5. This new has to be improved. Choose a more expressive type of graphic and use colors that are more distinctive

Response: Thank you very much for your and comments. We improved it in the latest version of the manuscript (Page 5-9).

  • Figure 6. This has to be displayed as a table, and this table has to be discussed in depth

Response: Thank you very much for your and comments. We revised it as a table in the latest version of the manuscript (Page 9).

Reviewer 2 Report

Manuscript entitled “Eucalyptus Leaf Solution Replace Heavy Metals to Removal of 2 Cyanobacteria in Paper Mill Industry Wastewater” submitted by Zhewei Hu, Shu Jin, Rongrong Ying, Xiaohui Yang and Baoping Sun, can be accepted for publication in Water Journal, after a serious major revision.

Here is a list of my specific comments:

  1. General comment 1: The novelty and practical applicability of this study should be clearly highlighted in the manuscript.
  2. General comment 2: The presentation of the experimental results is too brief. Pay attention on their interpretation in accordance with the main objectives of this study.
  3. Page 1, 1. Introduction: This section is too brief and should be detailed in order to describe the state of art in this field.
  4. Page 1, line 28: “Eutrophication is caused by excessive levels…”. Include here as reference the paper: Evolution of trophic parameters from Amara Lake, Environmental Engineering and Management Journal, 14(3), (2015), 559-565, because it is relevant for this observation.
  5. Page 2, line 54: “Although heavy metals can significantly…”. This paragraph should be reformulated because it is unclear.
  6. Page 2, line 64: “Therefore, the objectives…”. At the end of Introduction, the main objectives of this study should be clearly and detailed presented.
  7. Page 3, Figure 1: This figure should be deleted because it is irrelevant.
  8. Page 3, 2.2.2. Determination of the Physicochemical Properties of Water: More technical details should be added in this sub-section.
  9. Page 4, 3.1. Variations of Water Physicochemical Properties in Treatments: This section should be systematized. The experimental results included in this section should be logically presented and more detailed interpreted.
  10. Page 5, 3.2. Variations in Color and UV: Pay attention on the interpretation of the experimental results included in this section.
  11. Page 5, 3.3. Variations in Water Conductivity: The same observation as above.
  12. Page 6, 3.4. Variations in Total Dissolved Solids and Turbidity: The same observation.
  13. Page 8, 4. Conclusions: This section is too brief and should be detailed. Include here the most important experimental results and findings to highlight the importance of this study.

Author Response

Dear Editors and Reviewers:

Thank you for your letter and for the reviewers' comments concerning our manuscript entitled “Eucalyptus Leaf Solution Replace Metals to Removal of Cyanobacteria in Paper Mill Industry Wastewater” (No. water-1149172). Those comments are all valuable and very helpful for revising and improving our paper, as well as the important guiding significance to our researches. We have studied comments carefully and have made correction which we hope meet with approval. Besides, our current manuscript followed the journal formatting guidelines of Water. Revised portion are marked in red throughout the revised manuscript. The main corrections in the paper and the Responses to the reviewer’s comments are as flowing:

NOTE: All the Page and Line numbers where revisions were made refer to the Manuscript and Highlight with marked changes (Manuscript_revised version).docx. The Manuscript_Clean Version was the same version of the Manuscript_revised version with cleaned from all the marks.

Responses to the Reviewer #2’s comments:

  • Manuscript entitled “Eucalyptus Leaf Solution Replace Heavy Metals to Removal of 2 Cyanobacteria in Paper Mill Industry Wastewater” submitted by Zhewei Hu, Shu Jin, Rongrong Ying, Xiaohui Yang and Baoping Sun, can be accepted for publication in Water Journal, after a serious major revision.

 Response: Thank you very much for your comments and suggestions to revise the manuscript. All the Page and Line numbers where revisions were made refer to the Manuscript and Highlight with marked changes. We revised it in terms of the comments as follows:

  • General comment 1: The novelty and practical applicability of this study should be clearly highlighted in the manuscript.

Response: Thank you very much for your and comments. We improved it in the latest version of the manuscript (Page 2).

  • General comment 2: The presentation of the experimental results is too brief. Pay attention on their interpretation in accordance with the main objectives of this study.

Response: Thank you very much for your and comments. We improved it in the latest version of the manuscript (Page 4-10).

  • Page 1, 1. Introduction: This section is too brief and should be detailed in order to describe the state of art in this field.

Response: Thank you very much for your and comments. We improved it in the latest version of the manuscript (Page 2).

  • Page 1, line 28: “Eutrophication is caused by excessive levels…”. Include here as reference the paper: Evolution of trophic parameters from Amara Lake, Environmental Engineering and Management Journal, 14(3), (2015), 559-565, because it is relevant for this observation.

Response: Thank you very much for your and comments. We cited it in the latest version of the manuscript (Page 2).

  • Page 2, line 54: “Although heavy metals can significantly…”. This paragraph should be reformulated because it is unclear (Page 2).

Response: Thank you very much for your and comments. We improved it in the latest version of the manuscript.

  • Page 2, line 64: “Therefore, the objectives…”. At the end of Introduction, the main objectives of this study should be clearly and detailed presented (Page 2).

Response: Thank you very much for your and comments. We improved it in the latest version of the manuscript.

  • Page 3, Figure 1: This figure should be deleted because it is irrelevant.

Response: Thank you very much for your and comments. We think the figure 1 was important links in the process of experiment design and we reserved it in the latest version of the manuscript (Page 3).

  • Page 3, 2.2.2. Determination of the Physicochemical Properties of Water: More technical details should be added in this sub-section.

Response: Thank you very much for your and comments. We added the details in the latest version of the manuscript (Page 3-4).

  • Page 4, 3.1. Variations of Water Physicochemical Properties in Treatments: This section should be systematized. The experimental results included in this section should be logically presented and more detailed interpreted (Page 4-10).

Response: Thank you very much for your and comments. We improved it in the latest version of the manuscript (Page 4-10).

  • Page 5, 3.2. Variations in Color and UV: Pay attention on the interpretation of the experimental results included in this section.

Response: Thank you very much for your and comments. We improved it in the latest version of the manuscript (Page 4-10).

  • Page 5, 3.3. Variations in Water Conductivity: The same observation as above.

Response: Thank you very much for your and comments. We improved it in the latest version of the manuscript (Page 4-10).

  • Page 6, 3.4. Variations in Total Dissolved Solids and Turbidity: The same observation.

Response: Thank you very much for your and comments. We improved it in the latest version of the manuscript (Page 4-10).

  • Page 8, 4. Conclusions: This section is too brief and should be detailed. Include here the most important experimental results and findings to highlight the importance of this study.

Response: Thank you very much for your and comments. We improved it in the latest version of the manuscript (Page 11).

Round 2

Reviewer 2 Report

Manuscript entitled “Eucalyptus Leaf Solution Replace Heavy Metals to Removal of Cyanobacteria in Paper Mill Industry Wastewater” submitted by Zhewei Hu, Shu Jin, Rongrong Ying, Xiaohui Yang and Baoping Sun, can be accepted for publication in Water Journal, after a minor revision.

Here is a list of my specific comments:

  1. Page 2, line 116: Delete “The”.
  2. Page 2, Materials and Methods: The main characteristics of wastewater used for experimental should be included in this section.
  3. Page 3, Figure 1: This figure, if not deleted, should be rethought, because in this form is too simplistic. Also, make sure that the notations in the figure caption match those in the figure.
  4. Page 7, line 321: “However, at high concentration,…”. This paragraph should be reformulated because it is unclear.
  5. Page 7, line 325: “In our study, Zn2+ was more toxic…”. How can be explained this observation.
  6. Page 9, line 486: “…could inhibit the synthesis of chlorophyll, algae biomass was not affected.”. This paragraph should be reformulated because it is unclear.

Author Response

Dear Editors and Reviewers:

Thank you for your letter and for the reviewers' comments concerning our manuscript entitled “Eucalyptus Leaf Solution Replace Metals to Removal of Cyanobacteria in Paper Mill Industry Wastewater” (No. water-1149172). Those comments are all valuable and very helpful for revising and improving our paper, as well as the important guiding significance to our researches. We have studied comments carefully and have made correction which we hope meet with approval. Besides, our current manuscript followed the journal formatting guidelines of Water. Revised portion are marked in red throughout the revised manuscript. The main corrections in the paper and the Responses to the reviewer’s comments are as flowing:

NOTE: All the Page and Line numbers where revisions were made refer to the Manuscript and Highlight with marked changes (Manuscript_revised version).docx. The Manuscript_Clean Version was the same version of the Manuscript_revised version with cleaned from all the marks.
